# Using enriched semantic event chains to model human action prediction based on (minimal) spatial information

**Fatemeh Ziaeetabar**[1]*, **Jennifer Pomp**[2], **Stefan Pfeiffer**[1], **Nadiya El-Sourani**[2], **Ricarda I. Schubotz**[2], **Minija Tamosiunaite**[1,3], **Florentin Wörgötter**[1]

**1** Institute for Physics 3 - Biophysics and Bernstein Center for Computational Neuroscience (BCCN), University of Göttingen, Göttingen, Germany, **2** Department of Psychology, University of Münster, Münster, Germany, **3** Department of Informatics, Vytautas Magnus University, Kaunas, Lithuania

* fziaeetabar@gwdg.de

**Data Availability Statement:** The dataset which includes human participants results in the VR experiment has been uploaded as Supporting Information files. The "Human participants dataset"

## Abstract

Predicting other people's upcoming action is key to successful social interactions. Previous studies have started to disentangle the various sources of information that action observers exploit, including objects, movements, contextual cues and features regarding the acting person's identity. We here focus on the role of static and dynamic inter-object *spatial relations* that change during an action. We designed a virtual reality setup and tested recognition speed for ten different manipulation actions. Importantly, all objects had been abstracted by emulating them with cubes such that participants could not infer an action using object information. Instead, participants had to rely only on the limited information that comes from the changes in the spatial relations between the cubes. In spite of these constraints, participants were able to predict actions in, on average, less than 64% of the action's duration. Furthermore, we employed a computational model, the so-called enriched Semantic Event Chain (eSEC), which incorporates the information of different types of spatial relations: (a) objects' touching/untouching, (b) static spatial relations between objects and (c) dynamic spatial relations between objects during an action. Assuming the eSEC as an underlying model, we show, using information theoretical analysis, that humans mostly rely on a mixed-cue strategy when predicting actions. Machine-based action prediction is able to produce faster decisions based on individual cues. We argue that human strategy, though slower, may be particularly beneficial for prediction of natural and more complex actions with more variable or partial sources of information. Our findings contribute to the understanding of how individuals afford inferring observed actions' goals even before full goal accomplishment, and may open new avenues for building robots for conflict-free human-robot cooperation.

## 1 Introduction

Human beings excel at recognizing actions performed by others, and they do so even before the action goal has been effectively achieved [1, 2]. Thus, humans engage in action prediction.

folder includes a file named: "readme.txt" which has a brief explanation about this dataset. All these human analysis can be repeated with this data.

**Funding:** The research leading to these results has received funding from the German Research Foundation (DFG) grant WO388/13-1 and SCHU1439/8-1 as well as the European Community's H2020 Programme (Future and Emerging Technologies, FET) under grant agreement no. 732266, Plan4Act.

**Competing interests:** The authors have declared that no competing interests exist.

During this process, the brain activates a premotor-parietal network [3] that largely overlaps with the networks needed for action execution and action imagery [4]. Though in recent years, some progress has been made towards computationally more concrete models of the mechanisms and processes underlying action recognition [5], it still remains largely unresolved how the brain accomplishes this complex task. Prediction of actions can rely on different sources of information, including manipulated objects [6–9], contextual objects [10, 11], movements [12], context [13] and features regarding the actress or actor [14]. A major aim of ongoing research is to disentangle the respective contribution and relevance of these sources of information feeding human action prediction. Since these sources are largely confounded even in simple instances of natural action, the experimental approach has to fully control or to bluntly eliminate all potentially confounding sources that are not in the focus of empirical testing.

Against this backdrop, the present study addressed the relevance of spatial relations between the objects in an action scene. Previous studies comparing manipulation of appropriate objects (i.e., normal actions) with manipulations of inappropriate objects (i.e., pantomime) showed that brain activity during action observation was largely explained by processing of the actor's movements [15]. As a caveat, this finding may be explained by the particular movement-focused strategy subjects selected in this study where normal and pantomime actions were presented in intermixed succession. Other studies show that motion features are used by the brain to segment observed actions into meaningful segments and to update internal predictive models of the observed action [16, 17]. Correspondingly, individuals segment actions into consistent, meaningful chunks [18, 19], and intra-individually, they do so in a highly consistent manner, albeit high inter-individual variability [16]. It has been argued that the objective quality of these chunks is that within the continuous sequence, breakpoints may convey a higher amount of information than the remainder of the event. Nevertheless, this suggestion remains speculative as long as we do not find a way to objectively quantify the flow of information that the continuous stream of input provides. This objectification is hampered by the fact that time-continuous information is highly variable with regard to spatial and temporal characteristics differing between action exemplars. Moreover, object information is a confounding factor in natural actions. As exemplars of object classes, individual objects provide information about possible types of manipulation the observer has learned these objects to be associated with [6, 20, 21]. For instance, knives are mostly used for cutting. Hence, objects can efficiently restrict the number of actions that an action observer expect to occur [6]. Speculatively, humans may use a mixed strategy exploiting object as well as spatial information, and this strategy may be adapted to current constraints. For instance, spatial information and, specifically, spatial relations that are in the center of the current study may become more relevant when objects are difficult to recognize, e.g. when observing actions from a distance, in dim light or in case when actions are performed with objects or in environments not familiar to an observer, or when objects are used in an unconventional way.

In the present study, we sought to precisely analyse and objectify the way that humans exploit information about spatial relations during action prediction. Eliminating object and contextual (i.e., room, scene) information as confounding factors, we tested the hypothesis that spatial relations between objects can be exploited to successfully predict the outcome of actions before the action aim is fully accomplished.

As the basis for spatial relation calculation we use extended semantic event chains (eSEC), introduced in our previous work for action recognition in computer vision [22]. This approach allows us to determine a sequence of *discrete* spatial relations between different objects in the scene throughout the manipulation. These sequences were shown to allow action prediction in computer vision applications [23]. Three types of spatial relations are calculated: (1) object

touching vs. non-touching in the scene, (2) static spatial relations, like *above* or *around* and (3) dynamic spatial relations like *moving together* or *moving apart*.

The approach was developed based on previous assumptions on the importance of spatial relations in action recognition [24–28] and stands in contrast to action recognition and prediction methods based on time continuous information, like trajectories [29–32] or continuous action videos [33–35]. It also stands in contrast to the methods exploiting rich contextual information [36–40]. Here we consider that time continuous information is much disturbed by intra-class variability of the same action, e.g. see [41], thus it is not the best source for action prediction, while contextual information in the current study we consider as distractors as explained above.

The current study consisted of the following steps:

- Creating a virtual reality database containing ten different manipulation actions with multiple scenarios each.

- Conducting a behavioural experiment in which human participants engaged in action prediction in virtual reality for all scenarios, where prediction time and prediction accuracy were measured.

- Calculating three types of spatial relations using the eSEC model: (1) touching vs. non-touching relations, (2) static spatial relations and (3) dynamic spatial relations.

- Performing an information theoretical analysis to determine how participants used these three types of spatial relations for action prediction.

- Training an optimal (up to the learning accuracy) machine algorithm to predict an action using the relational information provided by the eSEC model.

- Comparing human to the optimal machine action prediction strategies based on spatial relations.

The paper is organized as follows: In Section 2 we describe both, the setup of the experiments as well as the main aspects of data analysis. Here we keep the description of machine methods intuitive to make the paper accessible to psychology-oriented readers; in Section 3 we provide and explain results of the current study, in Section 4 we evaluate our findings and define implications for future work. In the Appendix 5 we provide the details of the machine algorithms.

## 2 General experimental protocols and methods

A flow chart of our study is depicted in Fig 1. In the following we will briefly describe each box in the flow-chart.

### 2.1 Virtual reality videos

We designed a set of ten actions and created multiple virtual reality videos for each action. The ten actions were: *chop*, *cut*, *hide*, *uncover*, *put on top*, *take down*, *lay*, *push*, *shake*, and *stir*. All objects, including hand and tools, were represented by cubes of variable size and color to serve object-agnostic (except the hand) action recognition. The hand was always shown as a red cube (Fig 2). Scene arrangements and object trajectories varied in order to generate a wide diversity in the samples of each manipulation action type. For each of the ten action types, 30 sample scenarios were recorded by human demonstration. All action scenes included different arrangements of several cubes (including distractor cubes) to ensure that videos were

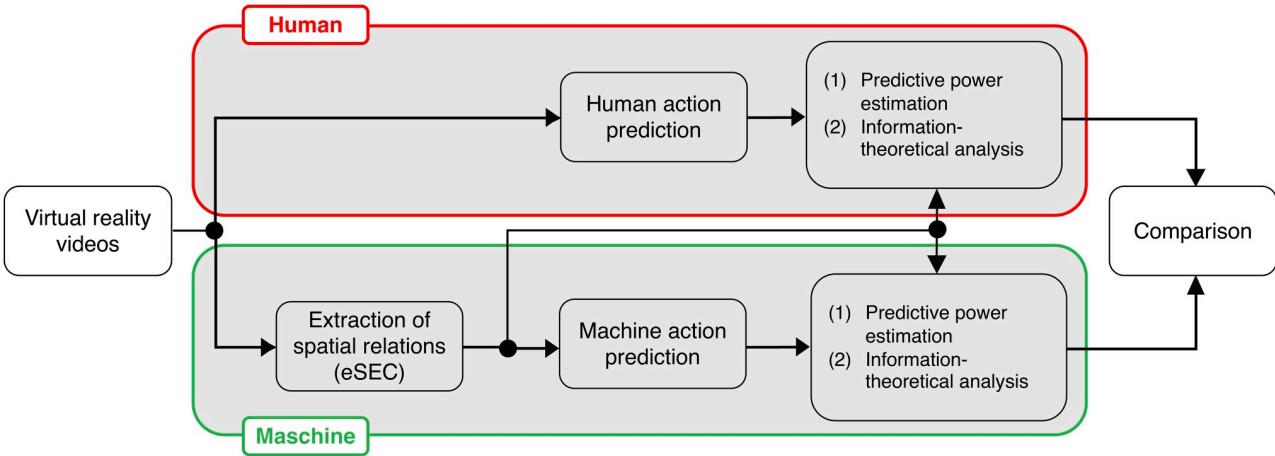

**Fig 1. Experimental schedule.**

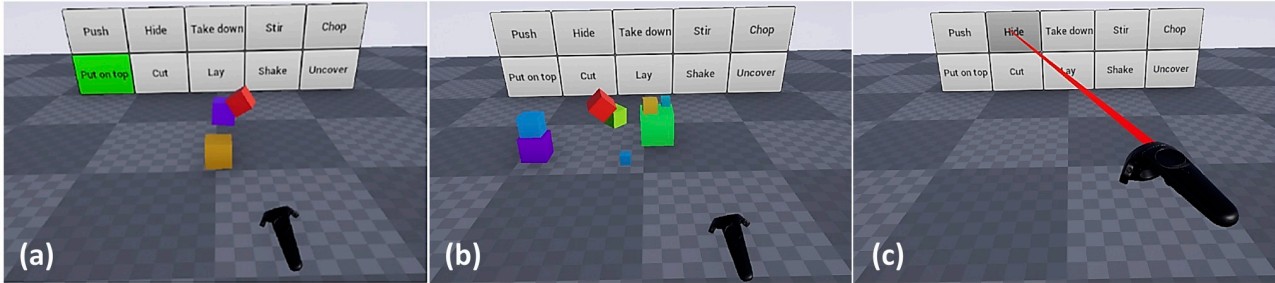

**Fig 2.** The VR experiment process, (a): experiment training stage for *put on top* action, (b): experiment testing stage: action scene playing and (c): experiment testing stage: selecting the action type.

indistinguishable at the beginning. The Virtual reality system as well as the ten actions mentioned above are specified in the Appendix, Subsections 5.1 to 5.3.

## 2.2 Behavioural study on action prediction

Forty-nine right-handed participants (20-68 yrs, mean 31.69 yrs, SD = 9.86, 14 female) took part in the experiment. One additional participant completed the experiments, but was excluded from further analyses due to an error rate of 14.7%, classified as outlier. Prior to the testing, written informed consent was obtained from all participants.

The experiment was not harmful and no sensitive data had been recorded and experimental data has been treated anonymously and only the instructions explained below had been given to the participants.

The experiment was performed in accordance with the ethical standards laid down by the 1964 Declaration of Helsinki. We followed the relevant guidelines of the Germany Psychological Society (Document: 28.09.2004 DPG: "Revision der auf die Forschung bezogenen ethischen Richtlinien") and also obtained official approval for these experiments by the Ethics Committee responsible at the University of Göttingen.

Participants were given a detailed explanation regarding the stimuli and the task of the experiment. They were then familiarized with the VR system and shown how to deliver their

responses during the experiment. The participants' task was to indicate as quickly as possible which action was currently presented.

Every experiment started with a short training phase in which one example of each action was presented. During this demo version, the name of the currently presented action was highlighted in green on the background board (see Fig 2(a)). After the training phase, we asked the participant if everything was clear and if he/she confirmed, we would start the test stage of the experiment.

During the test stage, a total of $30 \times 10$ action videos (trials) were shown to the participants in randomized order where the red hand-cube entered the scene and performed an action (Fig 2(b)). When the action was recognized and the participant pressed the motion controller's button, the moment of this button press was recorded as response time. Concurrently, all cubes disappeared from the scene so that no post-decision cogitation about the action was possible. At the same time, the controller was marked with a red pointer added to its front. Hovering over the action of choice and pressing motion controller's button again recorded the actual choice and advanced the experiment to the next trial (Fig 2(c)). Participants were allowed to rest during the experiment, and continued the experiment after resting. Since participants mostly proceeded quickly to the next trial, the overall duration of the experimental session usually did not exceed one hour. All experimental data were analysed using different statistical methods described in Subsections 2.5 and 2.6.

## 2.3 Extraction of spatial relations (eSEC)

The extended semantic event chain framework (eSEC) used as the underlying model in this study makes use of object-object relations. We defined three types of spatial relations in our framework: 1)"Touching" and "Non-touching" relations (TNR), 2) "Static Spatial Relation" (SSR) and 3)"Dynamic Spatial Relation" (DSR).

TNR between two objects were defined according to collision or "no collision" between their representative cubes.

SSR describe the relative position of two objects in space. We used the following SSRs: "Above", "Below", "Around", "Top", "Bottom", "AroundTouching", "Inside", "Surrounding" and "Null" (no relation in case two objects are too far away from each other). For algorithmic definition of those relations see Appendix, Subsection 5.5.

DSRs describe relative movements of two objects. We used the following DSRs: "Moving Together", "Halting Together" (describing the case where both objects are not moving), "Fixed-Moving Together" (describing the case when one object is moving across the other), "Getting Closer", "Moving Apart", "Stable" (describing the case when the distance between objects does not change), "No Relation" (describing the case when the distance between two objects exceeds a pre-defined threshold). For algorithmic definition of those relations see Appendix, Subsection 5.5.

Importantly, eSEC do not make use of any *real* object information. Objects remain abstracted (like in the VR experiments). We defined five abstract object types that play an essential role in any manipulation action and call them the **fundamental objects** (see Table 1). Fundamental objects 1, 2, and 3 *obtain* their role in the course of an action: they are numbered according to the order by which they encounter transitions between the relations N (non-touching) and T (touching). For example, 'fundamental object "1"' obtains its role given by "number 1" by being the first that encounters a change in touching (usually this is the object first touched by the hand).

Note that not all fundamental objects defined in Table 1 are always existing in a specific action. Only *hand*, *ground* and *fundamental object 1* are necessarily present in all analysed

**Table 1. Definition of the fundamental objects during a manipulation action [23].**

| Object | Definition | Remarks |
|---|---|---|
| **Hand** | The object that performs an action. | Not touching anything at the beginning and at the end of the action. It touches at least one object during an action. |
| **Ground** | The object that supports all other objects except the hand in the scene. | It is extracted as a ground plane in a visual scene. |
| **1** | The object that is the **first** to obtain a change in its T/N relations. | Trivially, the first transition will always be a touch by the hand. |
| **2** | The object that is the **second** to obtain a change in its T/N relations. | Either T→N or N→T relational change can happen. |
| **3** | The object that is the **third** to obtain a change in its T/N relations. | Either T→N or N→T relational change can happen. |

actions. The action-driven "birth" of objects 1, 2, and 3 automatically leads to the fact that irrelevant (distractor) objects are always ignored by the eSEC analysis.

Thus, the maximal number of relations that had to be analysed for an action was set by defined relations between fundamental objects: Given five object roles, there were $C(5, 2) = 10$ possible combinations leading to ten relations for each type (NTR, SSR, DSR), resulting in 30 relations in total.

The Enriched Semantic Event Chain (eSEC) is a matrix-form representation of the change of the three types of spatial relations described above throughout the action for the pairs of fundamental objects defined in Table 1. Fig 3 shows the eSEC matrix for a *put on top* action and demonstrates how relations change throughout this action.

## 2.4 Machine prediction

Machine prediction of a manipulation action was based on a learning procedure. For learning, we divided our data (eSEC tables) into *train* and *test* samples and performed a column-by-column comparison. That is, similarity values between the eSECs were derived by comparing each test action's eSEC (up to prediction column) to the every member of the training sample. We defined an action as "predicted" when the average similarity for one class remained high, while similarity for all other classes was low in this column. The similarity measurement algorithm between two eSEC matrices is explained in the Appendix, Subsection 5.6. Note, that the machine prediction algorithm, defined above, makes optimal action predictions based on eSEC information, to the precision of the applied learning procedure.

## 2.5 Comparison of human and machine predictive performance

We assessed predictive performance (of human or machine) relative to the length of the action measured in eSEC columns. The eSEC column at which prediction happens is called "prediction column". Predictive power is defined as:

$$P = (1 - \frac{column(\alpha)}{Total(\alpha)}) * 100\% \tag{1}$$

where $column(\alpha)$ is the "prediction column" and $Total(\alpha)$ is the total number of columns in the action $\alpha$ eSEC table. The earlier the action was predicted, the higher is the values of the measure $P$.

To compare human and machine predictive power, first, a repeated measures ANOVA on predictive power of humans were calculated with action (1—10) as within-subject factor. Then, human and machine performance was compared for each action separately using one-

## Static relations

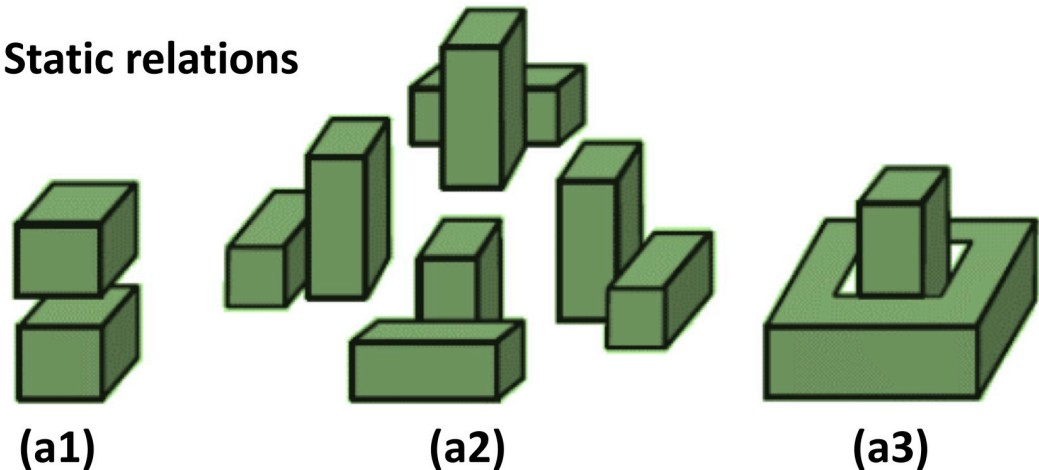

**(a1)** **(a2)** **(a3)**

## Dynamic relations

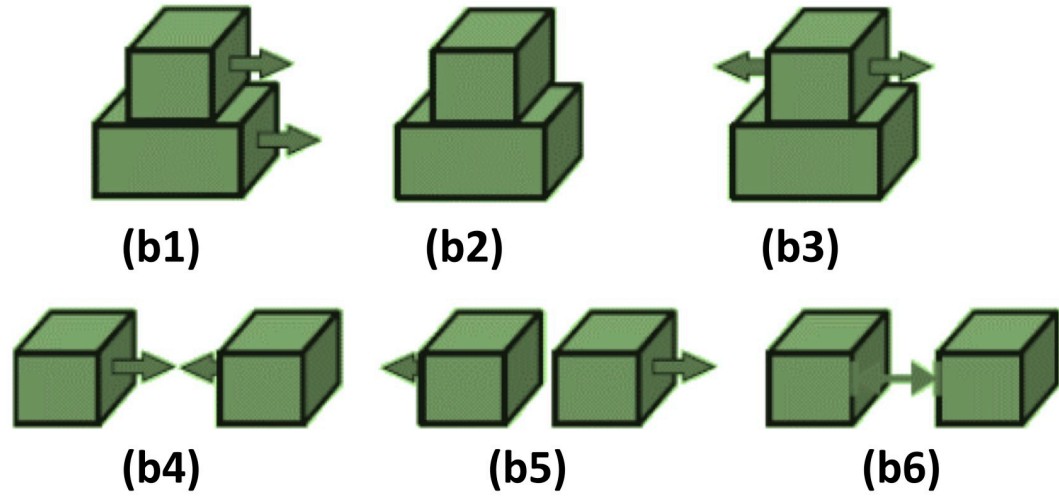

**(b1)** **(b2)** **(b3)**

**(b4)** **(b5)** **(b6)**

**Fig 3. Description of a "put on top" action in the eSEC framework with relation graph between all objects.** Only hand and ground are pre-specified, object 1 is the one first touched by the hand, object 2 the next where a touching/un-touching (T/N) change happens and object 3 in this case remains undefined (U) in all rows as there are no more T/N changes. This leads to the graph on the top left that shows all relations. Abbreviations in the eSEC are: U: undefined, T: touching, N: non-touching, O: very far (static), Q: very far (dynamic), Ab: above, To: top, Ar: around, ArT: around with touch, S: stable, HT: halt together, MT: move together, MA: moving apart, GC: getting close. Note that the two leftmost columns are identical for all actions as they indicate the starting situation before any action. The top, middle and bottom ten rows of the matrix indicate TNR, SSR and DSR between each pair of fundamental objects in a "put on top" action, respectively.

sample t-tests. As the machine data do not show variance, their predictive power value was used to compare it to human performance.

In addition, to inspect for the presence of learning effects in the human sample, correlations (Spearman Rho) were calculated for the number of trial (1—30) per action and predictive power as well as error rate.

Data were analysed using RStudio (Version 1.2.5001, RStudio Inc.) and SPSS 26 (IBM, New York, United States).

## 2.6 Information theoretical analysis

To model human action prediction based on eSEC matrices, we calculated the informational gain based on each eSEC column entry. More specifically, based on the eSEC descriptions of all ten actions, we derived a measurement of the amount of information presented in each column (or action step) of each action in comparison to all other actions. Each eSEC column, for a given sub-table (Touching = T, Static = S, Dynamic = D), contains ten coded descriptions of the spatial relations between hand, objects and ground. By stringing the eSEC codes of one column together, each column gets a new single code formally describing the action stage of a sub-table the participant observes at that moment. By taking the frequency of each action step or column-code across all 10 actions, we calculated the likelihood of a specific code in reference to the other actions in this column. So, if all eSEC descriptions are the same for one column, this column-code is assigned a likelihood of "1". If only one action differs (from the remaining nine actions), it gets a likelihood of 0.1 and the column-code of the differing action receives a likelihood value of 0.9, and so forth. Because not every action has the same number of columns, the lack of eSEC descriptions is also treated as a possible event. That means, if for example seven out of ten actions already have stopped at one point in time, these seven actions would receive a likelihood of 0.7 for this specific column.

We conducted this likelihood assignment procedure for each of the three types of information (TNR, SSR, DSR) separately. Note that the likelihood also gives an estimate of the information about one action that is presented in a column. If the likelihood of an action code is low, only a few or just this single action has this particular action code. So, if this code appears, it powerfully constrains action prediction.

Based on the likelihood $p$ of an action step $x$, we then calculated *bit rates* to quantify (self-) information I according to Shannon [42]:

$$I(x) = -log_2(p_x) \tag{2}$$

This transformation into information has two advantages over calculating with likelihoods. Firstly, it is more intuitive because more information is also displayed as a higher value, and secondly, we now were able to derive cumulated information by adding up the information values associated with successive columns. The transformation and cumulation were also done for all three information types separately. Thus, we obtained information values for each action step for each type of information separately. The additivity of the data also made it possible to combine multiple types of information by simply summing up the columns of the sub-tables.

Based on these information values, we modelled human performance. We employed the following models: one based only on TNR, one based only on SSR, one based only on DSR, three models adding two of the three types of information (T+S; T+D; S+D), one model adding all three types of information (T+S+D) and finally one model that ignores the three differing types of information and calculates the self-information based on all eSEC entries independent of the information type (Overall). For each model and for each action separately, a logistic regression was calculated using SPSS26. Each logistic regression included the absolute amount of information per action step according to the respective model, the accumulated information up to each action step, and the interaction term of these absolute and accumulated predictors. The logistic regressions' dependent binary variable was the presence of a response during the respective action step, indicating whether the action was predicted during this action step or not. Since predictors were correlated, models were estimated using the stepwise forward method for variable entry. Note that we did not interpret the coefficients and therefore did not need to regularize the regression model due to coefficient's correlation. Model fits were compared using the BIC (Bayesian-Information-Criterion) [43].

## 3 Results

In the human reaction time experiments, response times that exceeded the length of the action video were treated as time-outs and corresponding trials (13 out of 14700) were excluded from further analyses.

Participants' mean prediction accuracy was very high with a mean of 97.6% ($SD$ = 1.8%, $n$ = 49), ranging from 93.0% to 100.0%. Participants' mean predictive power ranged from 29.34 to 44.56 ($M$ = 37.03, $SD$ = 3.44, $n$ = 49). Regarding learning effects, hence, possible trends in performance change along an experiment, correlation analyses showed a significant reduction effect in error rates ($rs$ = −.72, $p$ <.001, $n$ = 30) and a significant enhancement effect for human predictive power ($rs$ = .96, $p$ <.001, $n$ = 30). Over trials, the mean error rate ranged from 0.004 to 0.063 ($M$ = 0.024, $SD$ = 0.015, $n$ = 30) and the mean predictive power ranged from 31.51 to 39.56 ($M$ = 37.00, $SD$ = 1.87, $n$ = 30).

Human predictive power was further analysed using a repeated measures ANOVA with action as within-subjects factor and, due to the significant learning effect, trial as second within-subject factor. Therefore, we pooled each six trials and used trial as a factor with five levels. Mauchly's test indicated that the assumption of sphericity was violated for action ($\chi^2(44)$ = 302.02, $p$ <.001), trial ($\chi^2(9)$ = 109.20, $p$ <.001) and for the interaction of action and trial ($\chi^2(665)$ = 1226.79, $p$ <.001), therefore degrees of freedom were corrected using Greenhouse-Geisser estimates of sphericity (action: $\epsilon$ = 0.37, trial: $\epsilon$ = 0.43, action × trial: $\epsilon$ = 0.40). The main effect of action was significant ($F(3.37, 161.78)$ = 427.96, $p < .001$, $\eta_p^2 = .899$) just as the main effect of trial ($F(1.71, 82.16)$ = 43.07, $p < .001$, $\eta_p^2 = .473$) and the interaction effect ($F(14.49, 695.46)$ = 2.95, $p < .001$, $\eta_p^2 = .058$). For further analysis each action was considered individually. As shown in Fig 4, predictive power varied strongly between actions. For instance, put and take actions were not correctly classified before most columns of the video (88% and 72%, respectively) were already presented, whereas cut, stir and uncover did only need about half (48%, 51%, and 52%, respectively) of the video time.

Separate one-sample t-tests per action for human vs. optimal (machine) predictive power consistently showed lower predictive power for the human ($ts$ < −2, $ps$ <.05). See details in Fig 4. Predictive power ranged from 14.3% to 62.5% for the machine, whereas humans predictive power ranged from 6.2% to 58.3%. On average, the machine spared observation of the remaining 45.6% of the video columns, humans the remaining 37%. In half of the actions (*take, uncover, hide, push* and *put on top*), this difference reached a very large effect size ($ds$ > 1). Interestingly, most pronounced differences not in terms of effect size but in terms of overall sampling time emerged for actions that were most quickly classified by the algorithm (take, uncover, cut). For take actions, humans sampled twice as many columns (72%) as the optimal performing algorithm (38%).

Logistic regressions revealed significant results for the eight models for each action respectively. All models tested significantly against their null model ($ps$ <.001). Fig 5 shows McFadden $R^2$ and BIC per model per action. Shaded cells indicate which model fits best human action prediction behaviour based on the BIC. Deploying the AIC (Akaike-Information-Criterion) yielded similar results.

As to the type of information exploited for prediction, we found marked differences between human and machine strategies. The machine behaviour was perfectly predicted by the biggest local gain in information, i.e., by transition into the column where the action code became unique for the respective action (Fig 6). For instance, when dynamic information was the first to provide perfect disambiguation between competing action models, the algorithm always followed this cue immediately (this was the case for *cut* and *hide*). Likewise, static information ruled machine behaviour for *push* and *lay*, reflecting the earliest possible point of

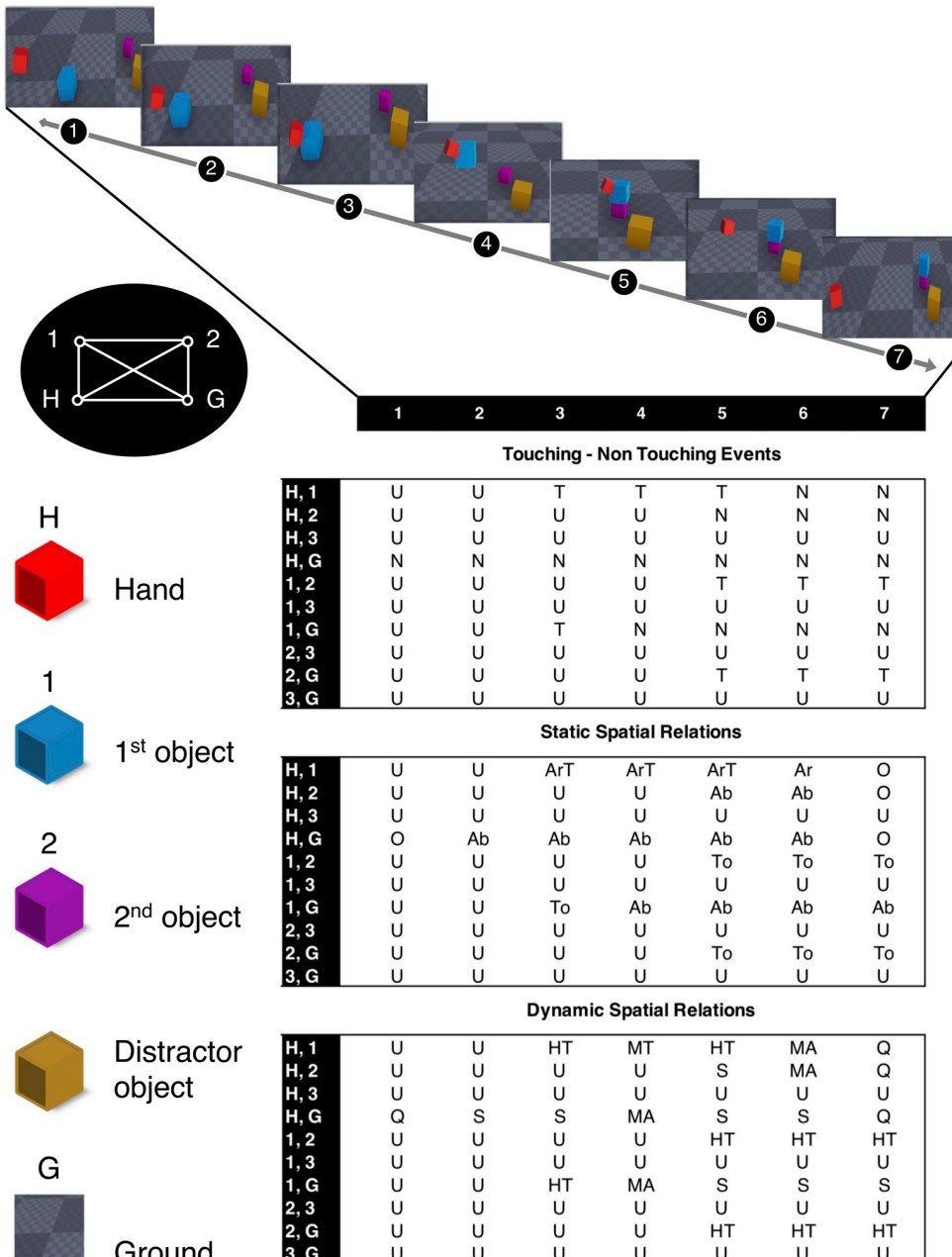

**Fig 4. Mean predictive power of human and machine.** t-values and p-values according to the t-tests per action.

certain prediction in these actions. Human suboptimal behaviour was nicely reflected by the fact (see Fig 5) that for *cut* and *hide*, subjects considered a combination of both dynamic and static spatial information (where they should have focused on dynamic information); the same strategy was applied to *push* and *lay*, where subjects should have better followed static information only.

Notably, when all three types of information (i.e., touching, static or dynamic information) were equally beneficial (this was the case fo*take, uncover, shake*, and *put*), human performance was best modelled by a combination of all three types of information (i.e., either T+S+D or

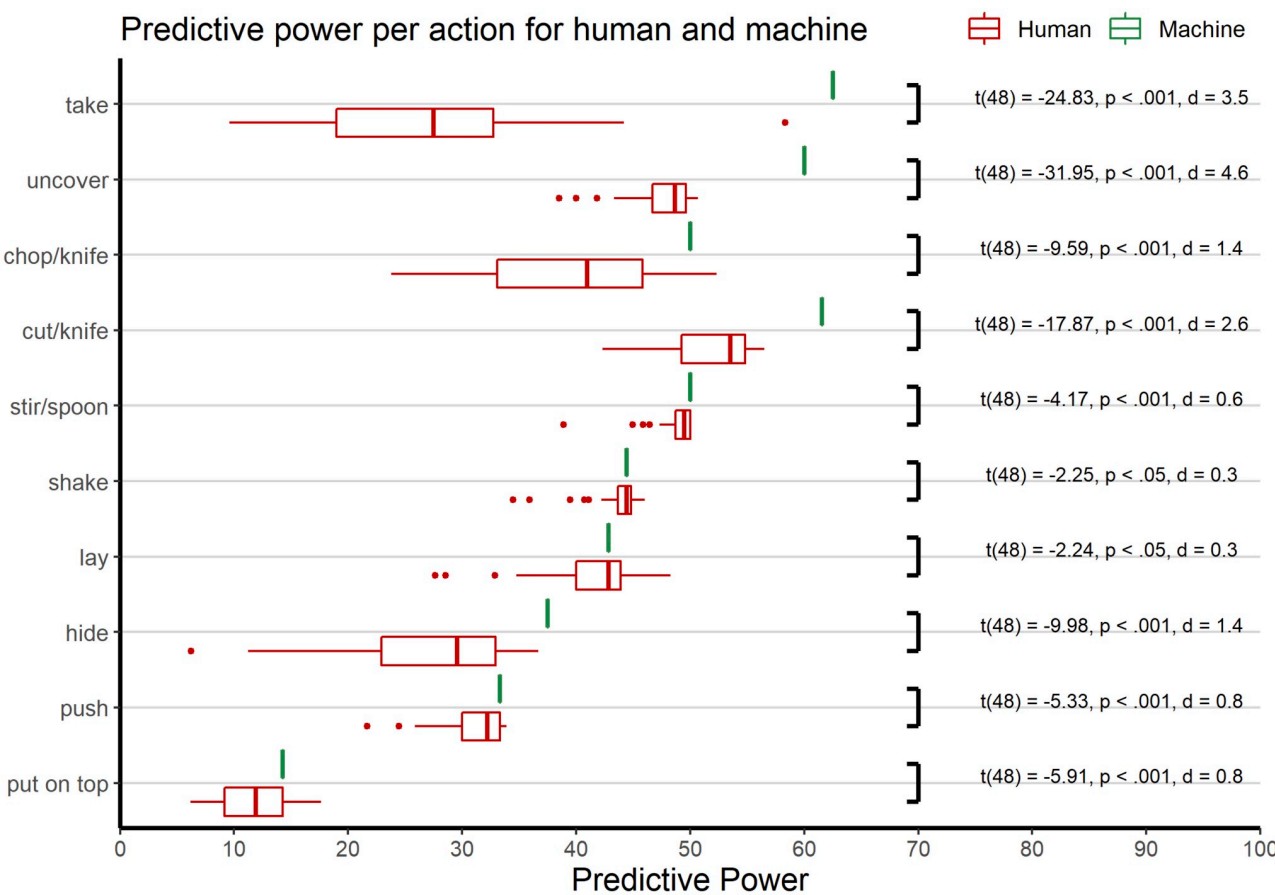

**Fig 5. Fitting different models to the actions.** (Abbreviations are shortened to allow to encode combinations by a short "+" annotation. We have Touch = T = TNR, Static = S = SSR, Dynamic = D = DSR. This leads to different combinations: T+S, T+D, S+D, T+S+D, where "Overall" refers to treating all eSEC columns independently of their individual information contents (see Methods).

Overall), with the exception of *chop*, where subjects followed static spatial information. A post-hoc paired-sample t-test showed a significant effect of *informational difference* ($t(48) = 15.95$, $p < .001$, $d_z = 2.3$). The z-transformed difference between mean human and machine predictive power was explicitly larger for informationally indifferent actions ($M = 2.1$) than for informationally different actions ($M = 1.1$). Expressed in non-transformed values, humans showed 12% less predictive power than the algorithm for informational indifferent action categories, but only 5% for the informational different ones.

## 4 Discussion

Humans predict actions based on different sources of information, but we know only little about how flexible these sources can be exploited in case that others are noisy or unavailable. Also, to better understand the respective contribution of these different sources of information, one has to avoid confounds and to properly control or eliminate alternative sources when focusing on one of them. In the present study, we tested how optimal human action prediction is when only static and dynamic spatial information is available. To this end we used action videos which were highly abstracted dynamic displays containing cubic place holders for all objects including hands, so that any information about real-world objects, environment, context, situation or actor were completely eliminated. We modelled human action prediction by

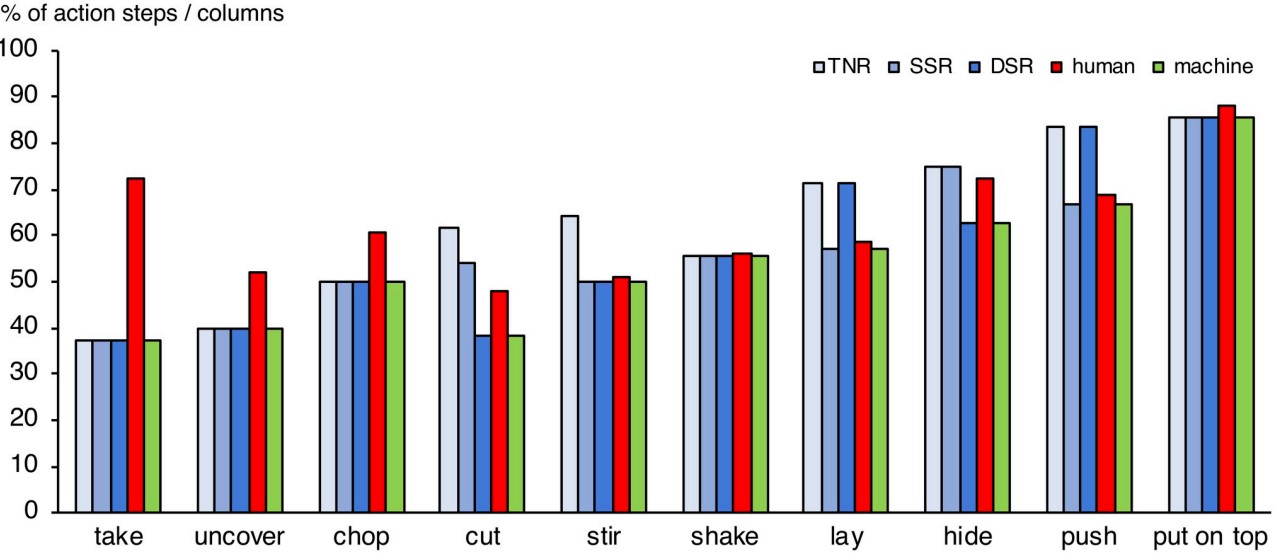

**Fig 6. Comparison of human (red bars) and machine (green bars) predictive performance.** Blue bars indicate the relative amount (percentage) of action steps elapsed per action, before the TNR (light blue), DSR (blue) or SSR (dark blue) model provided maximal local informational gain, enabling a secure prediction of the respective action. For instance, the 5th eSEC-column of the overall 13 eSEC-columns describing the *cut* action provided a unique description in terms of DSR. That is, after around 38% of these action's columns, the *cut* action could be predicted on the basis of DSR information, and this is what the algorithm did, as indicated by the green bar of equal length. In contrast, humans correctly predicted the *cut* action at the 6th (mean 6.26) column, corresponding to 48% of this action, exploiting both dynamic and static spatial information (cf. Fig 5 for this outcome).

an algorithm called enriched semantic event chain (eSEC), which had been derived from older "grammatical" approaches towards action encoding [24–26, 28, 44]. This algorithm is solely based on spatial information in terms of touching and untouching events between objects, their static and dynamic spatial relations.

Results show that participants performed strikingly well in predicting object-abstracted actions, i.e., by assigning the ongoing video to one out of ten basic action categories, before the video was completed. On average, they spared observation of the remaining 37% of each video. This finding suggests that humans engage in action prediction even on the basis of only static and dynamic spatial information, if other sources of information are missing.

Future studies have to examine how much real object or contextual information would further improve this performance level. Especially, object information provides an efficient restriction on to-be-expected manipulations [6–11]. It remains to be tested how non-spatial object information potentially interacts with the exploitation of static and dynamic spatial relations between objects involved in actions. Moreover, actions occur in certain contexts and environments that further restrict the observer's expectation, for instance with regard of certain classes of actions [2, 13, 45–47].

Albeit humans performed very well in action prediction, the machine algorithm, which was able to use the eSEC information optimally (up to the learning precision of that algorithm), consistently outperformed our participants, and this difference was significant for each single action category. On average, humans achieved about 91% of the predictive power of the machine. Based on an information theoretic approach, further analyses revealed that humans —in this particular setting—did not select the optimal strategy to disambiguate actions as fast as possible: While the machine reliably detected the earliest occurrence of disambiguation between the ongoing action and all other action categories, as indicated by the highest gain in information at the respective action step, human subjects did so in only half of the action categories. Instead, humans unswervingly applied a mixing strategy, concurrently relying on both

dynamic and spatial information in 8 out of 10 action types. This strategy was particularly disadvantageous for actions that were equally well predictable based on either static (NTR, SSR) or dynamic (DSR) information. Particularly in these—one may say—*informationally indifferent* cases, humans were significantly biased towards prolonged decisions: here, they showed 12% less predictive power than the algorithm as compared to 5% for the informationally different actions.

Summarizing these effects, the human bias towards using mixing strategies, combining static and dynamic spatial information, and to prolonged decisions for informational indifferent action categories establish overall poorer human predictive power. In principle, these two effects may result from the same general heuristics of human action observers, to exploiting multiple sources of information rather than relying on the first available source only. As a consequence, individuals prioritize correct over fast classification of observed actions.

Let us also note that the eSEC, which here was used to model human action recognition, is an advanced approach in the field of machine vision. It is finer grained and more expressive than its predecessor Semantic Event Chain [24, 44], but does not use fine (and inter-personally variable) spatial details, as compared to the Hidden Markov Model (HMM) algorithm, which, though being a classical approach, still represents the current state of the art in spatial-information (e.g. trajectory) based action recognition [48, 49]. In a previous study [23], we compared the predictive power of the eSEC framework with an HMM [32, 50]. The study was done on two real data sets, and we found that the average predictive power of eSECs was 61.9% as compared to only 32.4% for the HMM-based approach. This is because the three types of spatial relations, comprising the eSEC columns, capture important spatial and temporal properties of an action.

Describing action with a grammatical structure [25–27] such as eSEC [22, 23], renders a simple and fast framework for recognition and prediction in the presence of unknown objects and noise. This robustness lends itself to an intriguing hypothesis, which is asking to what degree such an event-based framework might help young infants to bootstrap action knowledge in view of the vast number of objects that they have never encountered before. In terms of spatial relations (as implemented in the current eSEC framework), the complexity of an action is far smaller than the complexity of the realm of objects with which an action can be performed, even when only considering a typical baby's environment. Clearly, this approach has proven to be beneficial for robotic applications [51] and we plan to extend it to complex actions and interactions between several agents (humans and robots) to examine the exploitation and exploration of predictive information during cooperation and competition.

## Limitations

Our approach did not take into account all dynamic and static spatial information provided by human action. For instance, we restricted dynamic spatial information to between-object change, whereas in natural action, we would also register dynamic within-hand change. Thus, actors shape their hands to fit the to-be-grasped object already when starting to reach out for it [52, 53], providing a valuable pointer to potentially upcoming manipulations and goals [54]. Likewise, gaze information plays a role in natural action observation [55], as the actors' looking to an object draws the observer's attention to the same object [56], and hence, potentially upcoming targets of the action.

Furthermore, our study was restricted to ten possible actions, whereas in everyday life, the number of potentially observable actions is much higher, resulting in higher uncertainty and higher competition among these potential actions. Speculatively, the human bias to employing mixed exploitation strategies may be better adapted to disambiguate actions among this

broader range of action classes. Future studies have to enlarge the sample of concurrently investigated actions to test this assumption and to increase overall ecological validity.

## 5 Appendix: Detailed methods

### 5.1 Virtual reality system

The main components of our VR system include computing power (for 3D data processing), head mounted display (for showing the VR content) and motion controllers (as the input devices). A Vive VR headset and motion controller released by HTC in April 2016 with a resolution of 1080 x 1200 per eye, have been used as our VR system. The "roomscale" system, which provides a precise 3D motion tracking between two infrared base stations, is the main advantage of this headset, which creates the opportunity to record and review actions for experiments on a larger scale of up to 5 meters diagonally. The Unreal Engine 4 (UE4) is a high performance game engine developed by Epic Games and is chosen as the game engine basis of this project. It has built-in support for VR environments and the Vives motion controllers.

### 5.2 Scenario recording

In order to make VR-videos for the 10 different actions, 30 variants of each action were recorded by two members of BCCN team (a 23 year old undergraduate male and a 30 year old doctoral student female). They implemented a VR platform by using C++ code structure. The motion controller is the core input component of the VR environment and they provided a separate function for each button on that by C++ programming. The designed system included three different modes. First, a mode to record new actions for the experiment; second, a mode to review in, and last, the experiment itself. To keep the controls as simple as possible and to avoid a second motion controller without implementing a complex physics system, the recording mode was split into two sub-modes: A single-cube recording mode (for single, mostly static cubes) and a two-cubes recording mode (for object manipulation).

### 5.3 Stimuli

Actions were defined as follows:

   *Chop*: The hand-object (hereafter: hand) touches an object (tool), picks up the object from the ground, puts it on another object (target) and starts chopping. When the target object has been divided into two parts, the tool object untouches the pieces of the target object. After that, the hand puts the tool object on the ground, untouches it, and leaves the scene.

   *Chop* scenarios had a **mean length** of **17.86** s (*SD* = **3.56, range = 13-27**).

   *Cut*: The hand touches an object (tool), picks up the object from the ground, puts it on another object (target) and starts cutting. When the target object was divided into two parts, the tool object untouches the pieces of the target object. After that, the hand puts the tool object on the ground, untouches it, and leaves the scene.

   *Cut* scenarios had a **mean length** of **19.50** s (*SD* = **3.13, range = 13-25**).

   *Hide*: The hand touches an object (tool), picks up the object from the ground, puts it on another object (target) and starts coming down on the target object until it covers that object thoroughly. Then the hand untouches the tool object and leaves the scene.

   *Hide* scenarios had a **mean length** of **13.43** s (*SD* = **2.40, range = 9-20**).

   *Uncover*: The hand touches an object (tool), picks up the object from the ground. The second object (target) emerges as the tool object is raised from the ground, because the tool object

had hidden the target object. After that, the hand puts the tool object on the ground, untouches it, and leaves the scene.

*Uncover* scenarios had a **mean length** of **12.66** s (*SD* = **3.20, range = 9-21**).

*Put on top*: The hand touches an object, picks up the object from the ground and puts it on another object. After that, the hand untouches the first object and leaves the scene.

*Put on top* scenarios had a **mean length** of **10.90** s (*SD* = **2.006, range = 8-16**).

*Take down*: The hand touches an object that is on another object, picks up the first object from the second object and puts it on the ground. After that, the hand untouches the first object and leaves the scene.

*Take down* scenarios had a **mean length** of **10.60** s (*SD* = **3.04, range = 6-18**).

*Lay*: The hand touches an object on the ground and changes its direction (lays it down) while it remains touching the ground. After that, the hand untouches the object and leaves the scene.

*Lay scenarios* had a **mean length** of **11.23** s (*SD* = **1.79, range = 8-15**).

*Push*: The hand touches an object on the ground and starts pushing it on the ground. After that, the hand untouches the object and leaves the scene.

*Push* scenarios had a **mean length** of **12.56** s (*SD* = **1.73, range = 9-17**).

*Shake*: The hand touches an object, picks up the object from the ground and starts shaking it. Then, the hand puts it back on the ground, untouches it and leaves the scene.

*Shake* scenarios had a **mean length** of **12.10** s (*SD* = **2.05, range = 9-17**).

*Stir*: The hand touches an object (tool), picks up the object from the ground, puts it on another object (target) and starts stirring. After that, the hand puts the tool object on the ground, untouches it, and leaves the scene.

*Stir* scenarios had a **mean length** of **20.23** s (*SD* = **4.67, range = 14-31**).

## 5.4 Details of machine action prediction

Note that all methodological details concerning our spatial relations definition (section 5.4.1) and their computation (section 5.5) as well as details of the similarity measurement algorithm (section 5.6) were reported previously in [23] and [28]. Hence, the next three subsections are essentially a repetition from those two papers without many changes.

**5.4.1 Spatial relations.**   The details on how to calculate static and dynamic spatial relations are provided below. Here we start first with a general description.

1. Touching and non-touching relations (TNR) between two objects were defined according to collision or non-collision between their representative cubes.

2. Static spatial relations (SSR) included: 'Above" (**Ab**), "Below" (**Be**), "Right" (**R**), "Left" (**L**), "Front" (**F**), "Back" (**Ba**), "Inside" (**In**), "Surround" (**Sa**). Since "Right", "Left", "Front" and "Back" depend on the viewpoint and directions of the camera axes, we combined them into "Around" (**Ar**) and used it at times when one object was surrounded by another. Moreover, "Above" (**Ab**), "Below" (**Be**) and "Around" (**Ar**) relations in combination with "Touching" were converted to "Top" (**To**), "Bottom" (**Bo**) and "Touching Around" (**ArT**), respectively, which corresponded to the same cases with physical contact. Fig 7 (a1-a3) shows static spatial relations between two objects cubes. If two objects were far from each other or did not have any of the above-mentioned relations, their static relation was considered as Null (**O**). This led to a set of nine static relations in the eSECs: **SSR** = {Ab, Be, Ar, Top, Bottom, ArT, In, Sa, O}. The additional relations, mentioned above: **R, L, F, Ba** are only used to define the relation Ar = around, because the former four relations are not view-point invariant.

| Action | | TNR | SSR | DSR | T+S | T+D | S+D | T+S+D | Overall |
|---|---|---|---|---|---|---|---|---|---|
| take | $R^2_{McF}$ | 0,099 | 0,099 | 0,099 | 0,099 | 0,099 | 0,099 | 0,099 | **0,164** |
|  | BIC | 7882,57 | 7882,57 | 7882,57 | 7882,57 | 7882,57 | 7882,57 | 7882,57 | **7323,09** |
| uncover | $R^2_{McF}$ | 0,486 | 0,486 | 0,277 | 0,486 | 0,541 | 0,541 | **0,543** | 0,532 |
|  | BIC | 4815,93 | 4815,93 | 6769,05 | 4815,93 | 4311,31 | 4311,31 | **4300,27** | 4399,94 |
| chop/knife | $R^2_{McF}$ | 0,162 | **0,198** | 0,182 | 0,177 | 0,172 | 0,19 | 0,18 | 0,147 |
|  | BIC | 8611,78 | **8245,08** | 8405,91 | 8462,61 | 8513,78 | 8324,06 | 8427,4 | 8765,91 |
| cut/knife | $R^2_{McF}$ | 0,331 | 0,498 | 0,501 | 0,42 | 0,436 | **0,501** | 0,466 | 0,28 |
|  | BIC | 6641,66 | 4994,44 | 4959,58 | 5757,74 | 5601,89 | **4957,43** | 5310,12 | 7130,39 |
| stir/spoon | $R^2_{McF}$ | 0,69 | 0,021 | **0,796** | 0,769 | 0,781 | 0,815 | 0,793 | 0,19 |
|  | BIC | 3167,84 | 9892,07 | **2087,19** | 2363,8 | 2246,09 | 1894,65 | 2115,6 | 8203,1 |
| shake | $R^2_{McF}$ | 0,733 | 0,756 | 0,733 | 0,756 | 0,733 | 0,756 | **0,757** | 0,701 |
|  | BIC | 2471,57 | 2273,84 | 2471,57 | 2267,71 | 2471,57 | 2267,71 | **2264,51** | 2772,08 |
| lay | $R^2_{McF}$ | 0,51 | 0,504 | 0,541 | 0,517 | 0,515 | **0,546** | 0,521 | 0,439 |
|  | BIC | 4129,8 | 4169,76 | 3870,66 | 4067,95 | 4082,34 | **3828,06** | 4038,63 | 4719,93 |
| hide | $R^2_{McF}$ | 0,333 | 0,333 | 0,331 | 0,333 | **0,334** | **0,334** | 0,334 | 0,107 |
|  | BIC | 5840,34 | 5840,34 | 5849,78 | 5840,34 | **5832,46** | **5832,46** | 5833,28 | 7792,97 |
| push | $R^2_{McF}$ | 0,741 | 0,695 | 0,741 | 0,741 | 0,741 | **0,741** | 0,741 | 0,695 |
|  | BIC | 2084,05 | 2438,16 | 2083,74 | 2084,02 | 2083,97 | **2083,46** | 2083,93 | 2438,16 |
| put on top | $R^2_{McF}$ | 0,482 | 0,482 | **0,482** | 0,482 | 0,482 | 0,482 | 0,482 | **0,482** |
|  | BIC | 4253,04 | 4253,04 | **4240,84** | 4253,04 | 4249,55 | 4249,55 | 4250,14 | **4240,84** |

**Fig 7.** (a) Static Spatial Relations: (a1) Above/Below, (a2) Around, (a3) Inside/Surround. (b) Dynamic Spatial Relations: (b1) Moving Together, (b2) Halting Together, (b3) Fixed-Moving Together, (b4) Getting Close, (b5) Moving Apart, (b6) Stable.

3. Dynamic Spatial Relations (DSR) require to make use of the frame history in the video. We used a history of 0.5 seconds, which is an estimate for the time that a human hand takes to change the relations between objects in manipulation actions. DSRs included the following relations: "Moving Together" (**MT**), "Halting Together" (**HT**), "Fixed-Moving Together" (**FMT**), "Getting Close" (**GC**), "Moving Apart" (**MA**) and "Stable" (**S**). DSRs between two objects cubes are shown in Fig 7 (b1-b6). MT, HT and FMT denote situations when two objects are touching each other while: both of them are moving in a same direction (MT), are motionless (HT), or when one object is fixed and does not move while the other one is moving on or across it (FMT). Case **S** denotes that any distance-change between objects remained below a defined threshold of $\xi = 1$ cm during the entire action. All these dynamic relations cases are clarified in Fig 7(b). In addition, **Q** is used as a dynamic relation between two objects when their distance exceeded the defined threshold $\xi$ or if they did not have any of the above-defined dynamic relations. Therefore, dynamic relations make a set of seven members: **DSR** = {MT, HT, FMT, GC, MA, S, Q}.

Finally, whenever one object became "Absent" or hidden during an action, the symbol (**A**) was used for annotating this condition. In addition, we use the symbol (**X**) whenever one object was destroyed or lost its primary shape (e.g. in *cut* or *chop* actions).

**5.4.2 Object types.** An exhaustive description of the five fundamental object types had been given in the main text and shall not be repeated here.

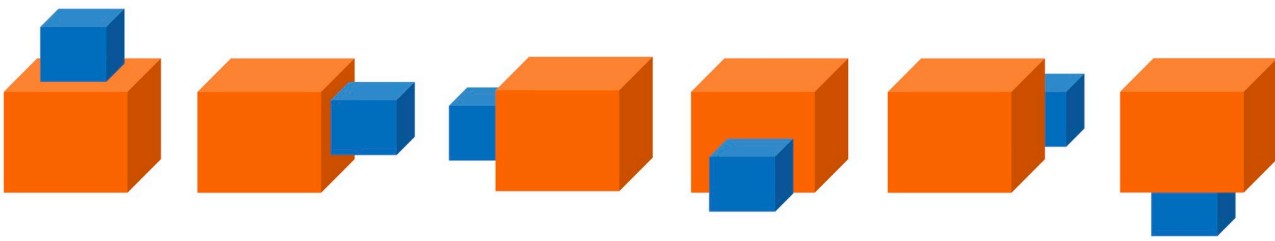

**Fig 8. Possible situations that two cubes touch each other.**

## 5.5 Mathematical definition of the spatial relations

As mentioned above, touching and non-touching relations between two objects are defined according to collision or non-collision between their representative cubes. 3D collision detection is a challenging topic which has been addressed in [57]. But, because the objects in our study are just cubes, we interpreted the contact of one of the six surfaces of one cube with one of the other cube's surfaces (see Fig 8) as touching event and this can be detected easily.

For example, in the left second situation of Fig 8, which has been shown with more details in Fig 9, the following condition will lead to a touching relation from a side.

$$
\begin{aligned}
& [x_1^\beta = x_1^\alpha] \\
\wedge \quad & [(y_1^\alpha < y_2^\beta < y_2^\alpha) \vee (y_1^\alpha < y_1^\beta < y_2^\alpha)] \\
\wedge \quad & [(z_1^\alpha < z_2^\beta < z_2^\alpha) \vee (z_1^\alpha < z_1^\beta < z_2^\alpha)]
\end{aligned}
\tag{3}
$$

Moreover, all discussed static and dynamic relations are defined by a set of rules. We start with explaining the rule set for static spatial relations and then proceed to dynamic spatial relations. In general, $x_{min}$, $x_{max}$, $y_{min}$, $y_{max}$, $z_{min}$ and $z_{max}$ indicate the minimum and maximum values between the points of object cube $\alpha_i$ in $x$, $y$ and $z$ axes, respectively.

Let us define the relation "Left", $SSR(\alpha_i, \alpha_j) = \mathbf{L}$ (object $\alpha_i$ is to the left of object $\alpha_j$) if:

$$
x_{max}(\alpha_i) < x_{max}(\alpha_j)
\tag{4}
$$

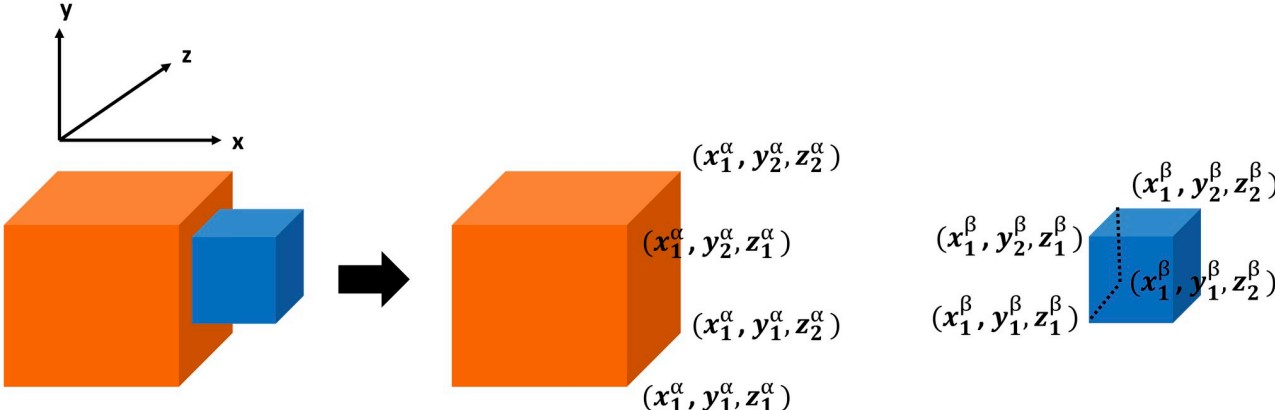

**Fig 9. Coordinate details of the two cubes that touch each other from side.**

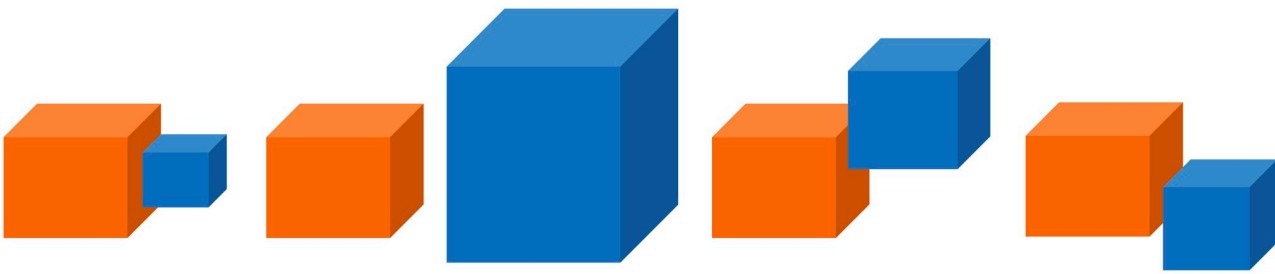

**Fig 10. Possible states of Left relation between two objects cubes when size and y positions vary.**

and the following exception condition holds

$$
\begin{aligned}
&[\neg(y_{min}(\alpha_i) > y_{max}(\alpha_j))] \\
\wedge\quad &[\neg(y_{min}(\alpha_j) > y_{max}(\alpha_i))] \\
\wedge\quad &[\neg(z_{min}(\alpha_i) > z_{max}(\alpha_j))] \\
\wedge\quad &[\neg(z_{min}(\alpha_j) > z_{max}(\alpha_i))]
\end{aligned}
\tag{5}
$$

The exception condition excludes from the relation "Left" those cases when two object cubes do not overlap in altitude (y direction) or front/back (z direction). Several examples of objects holding relation $SSR(red, blue) = \mathbf{L}$, when the size and shift in y direction varies, are shown in Fig 10.

$SSR(\alpha_i, \alpha_j) = \mathbf{R}$ is defined by $x_{max}(\alpha_i) > x_{min}(\alpha_j)$ and the identical set of exception conditions. The relations **Ab**, **Be**, **F**, **Ba** are defined in an analogous way. For **Ab** and **Be** the emphasis is on the "$y$" dimension, while for the **F**, **Ba** the emphasis is on the "$z$" dimension.

For the relation "inside" $SSR(\alpha_i, \alpha_j) = \mathbf{In}$ we use:

$$
\begin{aligned}
&[x_{min}(\alpha_j) \leq x_{min}(\alpha_i)] \wedge [x_{max}(\alpha_i) \leq x_{max}(\alpha_j)] \\
\wedge\quad &[z_{min}(\alpha_j) \leq z_{min}(\alpha_i)] \wedge [z_{max}(\alpha_i) \leq z_{max}(\alpha_j)] \\
\wedge\quad &[y_{min}(\alpha_j) \leq y_{max}(\alpha_i) \leq y_{max}(\alpha_j)]
\end{aligned}
\tag{6}
$$

The opposite holds for relation **Sa** (surrounding). For example, if $SSR(\alpha_i, \alpha_j) = \mathbf{In} \Rightarrow SSR(\alpha_j, \alpha_i) = \mathbf{Sa}$.

In addition of computing spatial relations TNR between two objects based on the above rules, we also check the touching relation between those two objects. This is then used to define several other relations. For example, if one object is above the other object, while they are touching each other, their static relation will be **To** (top).

$$
[SSR(\alpha_i, \alpha_j) = \mathbf{Ab}] \wedge [TNR(\alpha_i, \alpha_j) = \mathbf{T}] \Rightarrow [SSR(\alpha_i, \alpha_j) = \mathbf{To}]
\tag{7}
$$

There can be more than one static spatial relations between two object cubes. For example, one object can be both to the left and in back of the other object. However, to fill the eSEC matrix elements we need only one relation per object pair. This problem is solved by definition of a new notion called *shadow*.

Each cube has six surfaces. We label them as top, bottom, right, left, front and back based on their positions in our scene coordinate system. Whenever object $\alpha_i$ is to the left of object $\alpha_j$, one can make a projection from the right surface of object $\alpha_i$ onto the left rectangle of object $\alpha_j$

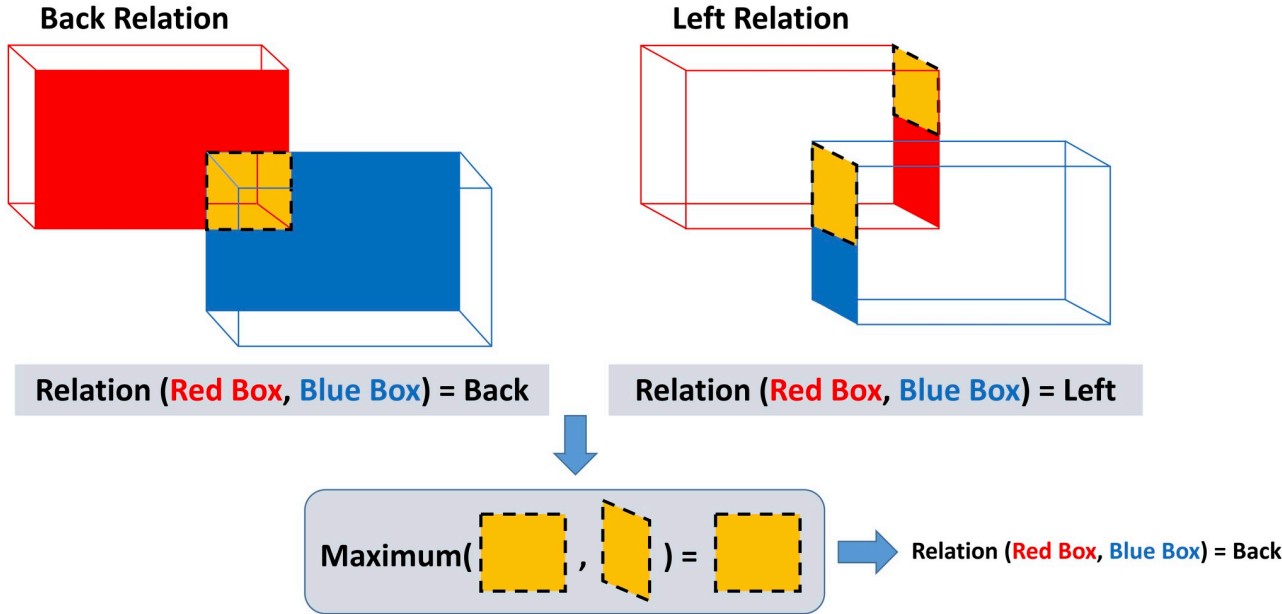

**Fig 11. Selection of one static spatial relation from several possible relations.**

and consider only the rectangle intersection area, This area is represented by the newly defined parameter *shadow*. Suppose $SSR(\alpha_i, \alpha_j) = \{R_1, \ldots, R_k\}$ while $R_1, \ldots, R_m \in SSR$ and we have calculated the $shadow(\alpha_i, \alpha_j, R)$ for all relations $R$ between the objects $\alpha_i$ and $\alpha_j$. The relation with the biggest shadow is then selected as the main static relation between the two objects: (Fig 11 includes the above description in the image format.)

$$SSR(\alpha_i, \alpha_j) = R_n (1 \leq n \leq k), if: \quad nonumber \tag{8}$$

$$shadow(\alpha_i, \alpha_j, R_n) = max_{1 \leq m \leq k}(Shadow(\alpha_i, \alpha_j, R_m)) \tag{9}$$

Dynamic spatial relations (DSR) are defined as follows. Suppose $Oi^f$ shows the central point of the object cube $\alpha_i^f$ (object $\alpha_i$ in $f_{th}$ frame); we define $\delta(\alpha_i^f, \alpha_j^f) = ||Oi^f - Oj^f||$ to be a two argument function for measuring the Euclidean distance between the cubes $\alpha_i$ and $\alpha_j$ in $f_{th}$ frame.

$$DSR(\alpha_i^f, \alpha_j^f) = \begin{cases} GC, & if \ \ \delta(\alpha_i^f, \alpha_j^f) - \delta(\alpha_i^{f+\theta}, \alpha_j^{f+\theta}) > \xi \\ MA, & if \ \ \delta(\alpha_i^{f+\theta}, \alpha_j^{f+\theta}) - \delta(\alpha_i^f, \alpha_j^f) > \xi \end{cases} \tag{10}$$

For this we use a time window of $\theta = 10$ frames (image snapshots in VR) in our experiments ($= 0.5s$); the threshold $\xi$ is kept at 0.1 m:

In the following we defined five conditions **P1** to **P5**, which then will be used to characterize the remaining DSRs.

$$
\begin{aligned}
\textbf{P1} : & \quad [TNR(\alpha_i^f, \alpha_j^f) = \textbf{T}] \wedge [TNR(\alpha_i^{f+\theta}, \alpha_j^{f+\theta}) = \textbf{T}] \\
\textbf{P2} : & \quad [TNR(\alpha_i^f, \alpha_j^f) = \textbf{N}] \wedge [TNR(\alpha_i^{f+\theta}, \alpha_j^{f+\theta}) = \textbf{N}] \\
\textbf{P3} : & \quad O_i^f \neq O_i^{f+\theta} \\
\textbf{P4} : & \quad O_j^f \neq O_j^{f+\theta} \\
\textbf{P5} : & \quad \delta(\alpha_i^{f+\theta}, \alpha_j^{f+\theta}) - \delta(\alpha_i^f, \alpha_j^f) < \xi
\end{aligned}
\tag{11}
$$

The dynamic relations **MT**, **HT**, **FMT** and **S**, based on the five conditions above are now defined in the following way:

$$
DSR(\alpha_i^f, \alpha_j^f) =
\begin{cases}
MT, & \text{if } P1 \wedge P3 \wedge P4 \\
HT, & \text{if } P1 \wedge \neg P3 \wedge \neg P4 \\
FMT, & \text{if } P1 \wedge (P3 \oplus P4) \\
S, & \text{if } P2 \wedge P5
\end{cases}
\tag{12}
$$

## 5.6 Similarity measure between eSECs

Suppose $\theta_1$ and $\theta_2$ are the names of two actions with eSECs that have $n$ and $m$ columns, respectively. We can concatenate the corresponding **TNR**, **SSR** and **DSR** of each fundamental object pair into a triple and make a 10-row matrix for $\theta_1$ and $\theta_2$ with ternary elements (TNR, SSR, DSR) instead of writing down a 30-row eSEC each:

$$
\theta_1 =
\begin{pmatrix}
(a_{1,1}, a_{11,1}, a_{21,1}) & (a_{1,2}, a_{11,2}, a_{21,2}) & \cdots & (a_{1,n}, a_{11,n}, a_{21,n}) \\
(a_{2,1}, a_{12,1}, a_{22,1}) & (a_{2,2}, a_{12,2}, a_{22,2}) & \cdots & (a_{2,n}, a_{12,n}, a_{22,n}) \\
\vdots & \vdots & \ddots & \vdots \\
(a_{10,1}, a_{20,1}, a_{30,1}) & (a_{10,2}, a_{20,2}, a_{30,2}) & \cdots & (a_{10,n}, a_{20,n}, a_{30,n})
\end{pmatrix}
$$

$$
\theta_2 =
\begin{pmatrix}
(b_{1,1}, b_{11,1}, b_{21,1}) & (b_{1,2}, b_{11,2}, b_{21,2}) & \cdots & (b_{1,n}, b_{11,n}, b_{21,n}) \\
(b_{2,1}, b_{12,1}, b_{22,1}) & (b_{2,2}, b_{12,2}, b_{22,2}) & \cdots & (b_{2,n}, b_{12,n}, b_{22,n}) \\
\vdots & \vdots & \ddots & \vdots \\
(b_{10,1}, b_{20,1}, b_{30,1}) & (b_{10,2}, b_{20,2}, b_{30,2}) & \cdots & (b_{10,n}, b_{20,n}, b_{30,n})
\end{pmatrix}
$$

We define the differences in the three different relation categories $L^{1:3}$, by using the elements of both matrices:

$$L^1_{i,j} = \begin{cases} 0, & \text{if } a_{i,j} = b_{i,j} \\ 1, & \text{otherwise} \end{cases}$$

$$L^2_{i,j} = \begin{cases} 0, & \text{if } a_{i+10,j} = b_{i+10,j} \\ 1, & \text{otherwise} \end{cases}$$

$$L^3_{i,j} = \begin{cases} 0, & \text{if } a_{i+20,j} = b_{i+20,j} \\ 1, & \text{otherwise} \end{cases}$$

where $1 \leq i \leq 10$, $1 \leq j \leq k$, $k = max(n, m)$.

Then the compound difference for the three categories is defined in the following way:

$$d_{i,j} = \frac{\sqrt{L^1_{i,j} + L^2_{i,j} + L^3_{i,j}}}{\sqrt{3}}. \tag{13}$$

If one matrix had more columns than the other matrix. i.e., $m < n$ or vice versa, the last column of the smaller matrix is repeated to match the number of columns of the bigger matrix. This leads to a consistent drop in similarity regardless of which two action are being compared.

Now we define $D$ as the matrix, which contains all compound differences between the elements of the two eSECs.

$$D_{(10,k)} = \begin{pmatrix} d_{1,1} & d_{1,2} & \cdots & d_{1,k} \\ d_{2,1} & d_{2,2} & \cdots & d_{2,k} \\ \vdots & \vdots & \ddots & \vdots \\ d_{10,1} & d_{10,2} & \cdots & d_{10,k} \end{pmatrix}$$

where $d_{i,j}$ denotes the dissimilarity of $i_{th}$ objects pair at the $j_{th}$ time stamp (column). Then, $D$, which is the total dissimilarity between eSECs of $\theta_1$ and $\theta_2$ is considered as the average across all elements of matrix $D$.

$$D_{\theta_1,\theta_2} = \frac{1}{k * 10} \left( \sum_{j=1}^{k} \sum_{i=1}^{10} d_{i,j} \right) \tag{14}$$

Accordingly, the *similarity* between these eSECs $Sim_{\theta_1,\theta_2}$, is measured as:

$$Sim_{\theta_1,\theta_2} = (1 - D_{\theta_1,\theta_2}) * 100\% \tag{15}$$

## Supporting information

**S1 Video.**
(MP4)

**S1 Dataset.**
(RAR)

## Author Contributions

**Conceptualization:** Fatemeh Ziaeetabar, Ricarda I. Schubotz, Minija Tamosiunaite, Florentin Wörgötter.

**Data curation:** Stefan Pfeiffer.

**Formal analysis:** Fatemeh Ziaeetabar, Jennifer Pomp, Nadiya El-Sourani.

**Funding acquisition:** Ricarda I. Schubotz, Florentin Wörgötter.

**Investigation:** Fatemeh Ziaeetabar, Stefan Pfeiffer.

**Methodology:** Fatemeh Ziaeetabar, Jennifer Pomp, Nadiya El-Sourani, Ricarda I. Schubotz, Minija Tamosiunaite, Florentin Wörgötter.

**Project administration:** Ricarda I. Schubotz, Minija Tamosiunaite, Florentin Wörgötter.

**Resources:** Florentin Wörgötter.

**Software:** Stefan Pfeiffer.

**Supervision:** Ricarda I. Schubotz, Minija Tamosiunaite, Florentin Wörgötter.

**Visualization:** Fatemeh Ziaeetabar, Jennifer Pomp, Stefan Pfeiffer.

**Writing – original draft:** Fatemeh Ziaeetabar, Jennifer Pomp, Ricarda I. Schubotz, Minija Tamosiunaite.

**Writing – review & editing:** Ricarda I. Schubotz, Minija Tamosiunaite, Florentin Wörgötter.

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
