## [Decision Letter · Decision Letter 0]

24 Aug 2020

PONE-D-20-16126

Human and Machine Action Prediction Independent of Object Information

PLOS ONE

Dear Dr. Ziaeetabar,

Thank you for submitting your manuscript to PLOS ONE. After careful consideration, we feel that it has merit but does not fully meet PLOS ONE’s publication criteria as it currently stands. Therefore, we invite you to submit a revised version of the manuscript that addresses the points raised during the review process.

We look forward to receiving your revised manuscript.

Kind regards,

Chen Zonghai

Academic Editor

PLOS ONE

Journal Requirements:

4. Please update your submission to use the PLOS LaTeX template. The template and more information on our requirements for LaTeX submissions can be found at http://journals.plos.org/plosone/s/latex.

Additional Editor Comments (if provided):

Revise according to the reviewer's opinion.

Reviewer 1

The manuscript entitled “Human and Machine Action Prediction Independent of Object Information” explores action prediction algorithms under no context condition. A virtual reality setup is established to research action recognition mechanism differences between human and machine vision. In manipulation actions, all objects are emulated with cubes so that human participants cannot infer action through object context and use spatial relations instead. Results show that participants are able to predict actions in, on average, less than 64% of the action's duration. In comparison, a computational model, the so-called enriched Semantic Event Chain (eSEC), which incorporates the information of spatial relations is employed. After being trained by the same actions as those observed by participants, this model successfully predicted actions even better than humans. Using information theoretical analysis, eSECs are able to make optimal use of individual cues, whereas humans seem to mostly rely on a mixed-cue strategy, which takes longer until recognition.

The research work reveals interesting mechanism of action prediction in human through well-designed comparison experiments. Providing a better cognitive basis of action recognition may, on the one hand improve our understanding of related human pathologies and, on the other hand, also help in building robots for conflict-free human-robot cooperation.

However, it remains to be promoted in following aspects:

1. What’s the motivation of this research? It should be stated in the beginning.

2. From introduction section, the necessity of human action prediction research without context information is not explained. This may benefit human-computer interaction, but in most applications, context information is available and is effective for action prediction.

3. The purpose of the research work is unclear. To explain human’s action prediction mechanism without context information or to propose a better action prediction algorithm? Experiments setup varies for different research purpose.

4. Section 1.2 should focus more on action prediction as it is the topic of this research.

5. Some details on human experiments should be clarified. In the short training phase, how to determine the end of training? Is it decided by researchers or participants? As it’s not a routine scenario, to make a fair comparison with machine vision, it should be decided by participants and an additional test should be added to validate that participants have been well-trained.

6. Are participants informed that their response time will be recorded as an evaluation criterion, which may affect their prediction timing?

7. What about prediction accuracy? Are all prediction results correct? How to analyze wrong predictions?

8. A typo mistake in line 140: two “for example”.(less...)

Reviewer 2

This paper proposes a system about machine based action recognition system eSEC learning and designed a virtual reality setup and tested recognition speed for different manipulation actions. The authors introduce in details how the theoretical analysis is done and recognition speed is performed.

Paper is not well organized and has limited potential for acceptance in “PLOS ONE”, in current format though there are some observations, corrections and suggestions regarding this paper.

• Author MUST clearly describe their contribution. Put another section what is author contribution?

• Separate introduction and literature review.

• Proposed work section is quite weak and needs major improvement. It lacks any flow diagram, algorithm, pseudo code etc. Each step of proposed algorithm/work should be clearly depicted how your work is different from existing work.

• Diagrams and flow charts are not good need to redraw.

• Performance measures should be more. The proposed work should be evaluated with a number of performance measures to prove its validity.

• Abstract and Conclusion are poorly written need much revision.

• Add references that are more recent.

• Overall, the paper lies in the category of revision.

Overall the language is not very good; however, it MUST be proofread if again before submission again.(less...)

Reviewer 3

Action prediction independent of object information as an observation and hypothesis is validated through a psycho-physical experiments rigorously conducted by the authors. A set of 10 actions are considered over a VR based experimental system. The authors further validated an eSEC computational framework to show that with eSEC, machine could achieve action prediction capability. The machine prediction power vs human prediction performance as a comparison is provided by the authors and some speculative explanations are given and discussed.

The draft is fairly well written and flows well. I really enjoyed reading the draft.

The experiments are thorough enough to approach their conclusions in my view.

The eSECs as a formulation and representation is adopted as a computational tool in this draft, is an appropriate choice given its prior use in similar computational problem domains (in robotics and computer vision fields).

The relevant literature is also well presented and reviewed.

Some parts of the draft could be improved by making the description more clear to the readers. For example, line 396 "when all three types of information"

It is unclear to me what is the third type of information other than dynamic and static ones? Please clarify.

Also, an interesting future question and direction could be, as most of the action recognition dataset and benchmarks in computer vision research area come up with the set of actions in a kind of ad-hoc manner (especially for manipulation action dataset). I would be keen to see the authors based on their discoveries from this draft to provide some designing principles for future action recognition dataset and challenges, that could fully consider the types of information discussed here.(less...)

Reviewer 4

Following are some observations

• Abstract is too much lengthy.

• Abstract is not written according to the theme of abstract.

• Actual methodology/algorithms are not mentioned in the abstract.

• In introduction section, contributions should be mentioned in bullets for better understanding of the readers.

• The manuscript should be checked for typos. In some places the word Figure is written while in other places Fig is written, must be uniform throughout.

• Figures quality is not good, must be 300dpi.

• Authors employed so-called extended semantic event chains (eSEC) which is an existing work, what is their real contribution?(less...)

Reviewers' comments:

Reviewer's Responses to Questions

**Comments to the Author**

1. Is the manuscript technically sound, and do the data support the conclusions?

Reviewer #1: Yes

Reviewer #2: Yes

Reviewer #3: Yes

Reviewer #4: Yes

2. Has the statistical analysis been performed appropriately and rigorously? 

Reviewer #1: Yes

Reviewer #2: No

Reviewer #3: Yes

Reviewer #4: I Don't Know

3. Have the authors made all data underlying the findings in their manuscript fully available?

Reviewer #1: Yes

Reviewer #2: Yes

Reviewer #3: Yes

Reviewer #4: Yes

4. Is the manuscript presented in an intelligible fashion and written in standard English?

Reviewer #1: Yes

Reviewer #2: No

Reviewer #3: Yes

Reviewer #4: Yes

5. Review Comments to the Author

Reviewer #1: The manuscript entitled “Human and Machine Action Prediction Independent of Object Information” explores action prediction algorithms under no context condition. A virtual reality setup is established to research action recognition mechanism differences between human and machine vision. In manipulation actions, all objects are emulated with cubes so that human participants cannot infer action through object context and use spatial relations instead. Results show that participants are able to predict actions in, on average, less than 64% of the action's duration. In comparison, a computational model, the so-called enriched Semantic Event Chain (eSEC), which incorporates the information of spatial relations is employed. After being trained by the same actions as those observed by participants, this model successfully predicted actions even better than humans. Using information theoretical analysis, eSECs are able to make optimal use of individual cues, whereas humans seem to mostly rely on a mixed-cue strategy, which takes longer until recognition.

The research work reveals interesting mechanism of action prediction in human through well-designed comparison experiments. Providing a better cognitive basis of action recognition may, on the one hand improve our understanding of related human pathologies and, on the other hand, also help in building robots for conflict-free human-robot cooperation.

However, it remains to be promoted in following aspects:

1. What’s the motivation of this research? It should be stated in the beginning.

2. From introduction section, the necessity of human action prediction research without context information is not explained. This may benefit human-computer interaction, but in most applications, context information is available and is effective for action prediction.

3. The purpose of the research work is unclear. To explain human’s action prediction mechanism without context information or to propose a better action prediction algorithm? Experiments setup varies for different research purpose.

4. Section 1.2 should focus more on action prediction as it is the topic of this research.

5. Some details on human experiments should be clarified. In the short training phase, how to determine the end of training? Is it decided by researchers or participants? As it’s not a routine scenario, to make a fair comparison with machine vision, it should be decided by participants and an additional test should be added to validate that participants have been well-trained.

6. Are participants informed that their response time will be recorded as an evaluation criterion, which may affect their prediction timing?

7. What about prediction accuracy? Are all prediction results correct? How to analyze wrong predictions?

8. A typo mistake in line 140: two “for example”.

Reviewer #2: This paper proposes a system about machine based action recognition system eSEC learning and designed a virtual reality setup and tested recognition speed for different manipulation actions. The authors introduce in details how the theoretical analysis is done and recognition speed is performed.

Paper is not well organized and has limited potential for acceptance in “PLOS ONE”, in current format though there are some observations, corrections and suggestions regarding this paper.

• Author MUST clearly describe their contribution. Put another section what is author contribution?

• Separate introduction and literature review.

• Proposed work section is quite weak and needs major improvement. It lacks any flow diagram, algorithm, pseudo code etc. Each step of proposed algorithm/work should be clearly depicted how your work is different from existing work.

• Diagrams and flow charts are not good need to redraw.

• Performance measures should be more. The proposed work should be evaluated with a number of performance measures to prove its validity.

• Abstract and Conclusion are poorly written need much revision.

• Add references that are more recent.

• Overall, the paper lies in the category of revision.

Overall the language is not very good; however, it MUST be proofread if again before submission again.

Reviewer #3: Action prediction independent of object information as an observation and hypothesis is validated through a psycho-physical experiments rigorously conducted by the authors. A set of 10 actions are considered over a VR based experimental system. The authors further validated an eSEC computational framework to show that with eSEC, machine could achieve action prediction capability. The machine prediction power vs human prediction performance as a comparison is provided by the authors and some speculative explanations are given and discussed.

The draft is fairly well written and flows well. I really enjoyed reading the draft.

The experiments are thorough enough to approach their conclusions in my view.

The eSECs as a formulation and representation is adopted as a computational tool in this draft, is an appropriate choice given its prior use in similar computational problem domains (in robotics and computer vision fields).

The relevant literature is also well presented and reviewed.

Some parts of the draft could be improved by making the description more clear to the readers. For example, line 396 "when all three types of information"

It is unclear to me what is the third type of information other than dynamic and static ones? Please clarify.

Also, an interesting future question and direction could be, as most of the action recognition dataset and benchmarks in computer vision research area come up with the set of actions in a kind of ad-hoc manner (especially for manipulation action dataset). I would be keen to see the authors based on their discoveries from this draft to provide some designing principles for future action recognition dataset and challenges, that could fully consider the types of information discussed here.

Reviewer #4: Following are some observations

• Abstract is too much lengthy.

• Abstract is not written according to the theme of abstract.

• Actual methodology/algorithms are not mentioned in the abstract.

• In introduction section, contributions should be mentioned in bullets for better understanding of the readers.

• The manuscript should be checked for typos. In some places the word Figure is written while in other places Fig is written, must be uniform throughout.

• Figures quality is not good, must be 300dpi.

• Authors employed so-called extended semantic event chains (eSEC) which is an existing work, what is their real contribution?

6. PLOS authors have the option to publish the peer review history of their article (what does this mean?). If published, this will include your full peer review and any attached files.

Reviewer #1: No

Reviewer #2: No

Reviewer #3: **Yes: **Yezhou Yang

Reviewer #4: No

---

## [Author Response · Author response to Decision Letter 0]

29 Oct 2020

First we want to thank the reviewers for the helpful comments. Further, we provide the answers to those comments. Reviewer comments will be written in Courier new font, our answers in Calibri font.

Please, note, that the figures in the merged submission PDF are not reproduced with good quality, but the PLOS guarantees the download of the high-resolution figures, see https://everyone.plos.org/2011/02/11/ask-everyone-figure-files-in-your-merged-pdf/

*****Reviewer 1*****

The manuscript entitled “Human and Machine Action Prediction Independent of Object Information” explores action prediction algorithms under no context condition. A virtual reality setup is established to research action recognition mechanism differences between human and machine vision. In manipulation actions, all objects are emulated with cubes so that human participants cannot infer action through object context and use spatial relations instead. Results show that participants are able to predict actions in, on average, less than 64% of the action's duration. In comparison, a computational model, the so-called enriched Semantic Event Chain (eSEC), which incorporates the information of spatial relations is employed. After being trained by the same actions as those observed by participants, this model successfully predicted actions even better than humans. Using information theoretical analysis, eSECs are able to make optimal use of individual cues, whereas humans seem to mostly rely on a mixed-cue strategy, which takes longer until recognition. The research work reveals interesting mechanism of action prediction in human through well-designed comparison experiments. Providing a better cognitive basis of action recognition may, on the one hand improve our understanding of related human pathologies and, on the other hand, also help in building robots for conflict-free human-robot cooperation. However, it remains to be promoted in following aspects:

1) What’s the motivation of this research? It should be stated in the beginning.

The motivation of the study was to find out whether and how humans use spatial relations between objects in prediction of manipulation actions. In the revised manuscript, we write in the Introduction (lines 83-88):

“In the present study, we sought to precisely analyze and objectify the way that humans exploit information about spatial relations during action prediction. Eliminating object and contextual (i.e., room, scene) information as confounding factors, we tested the hypothesis that spatial relations between objects can be exploited to successfully predict the outcome of actions before the action aim is fully accomplished.”

2) From introduction section, the necessity of human action prediction research without context information is not explained. This may benefit human-computer interaction, but in most applications, context information is available and is effective for action prediction. 

Our study is the first to specifically address how humans use relational spatial information for prediction. Furthermore, we write now (see lines 78-82): “For instance, spatial information and, specifically, spatial relations that are in the center of the current study may become more relevant when objects are difficult to recognize, e.g. when observing actions from a distance, in dim light or in case when actions are performed with objects or in environments not familiar to an observer, or when objects are used in an unconventional way.”

In addition, we had suggested in the discussion of the 1st submission already that infants could potentially bootstrap action understanding starting with spatial relations, at the stages of development where their general world knowledge is still limited and thus, contextual information cannot be interpreted in the same way by them as by a grown-up person. (See lines 458-465). 

3) The purpose of the research work is unclear. To explain human’s action prediction mechanism without context information or to propose a better action prediction algorithm? Experiments setup varies for different research purpose. 

The purpose of the research work is to investigate the usage of spatial relations for manipulation prediction by humans. Calculation of the spatial relations is based on the eSEC model, which was originally developed for computer vision. We now have much rewritten the introduction and method section to make it clear that we concentrate on explaining human’s action prediction mechanism. 

We also have changed the title of the paper to make the purpose of the paper more clear. The new title is: “Using enriched Semantic Event Chains to model human action prediction based on (minimal) spatial information”. 

4) Section 1.2 should focus more on action prediction as it is the topic of this research. 

Having refocused the paper on human action prediction, the introduction is not divided into sections any more. We have rewritten and much shortened the text from section 1.2, focusing at prediction as requested by the reviewer (see lines 98-106).

5) Some details on human experiments should be clarified. In the short training phase, how to determine the end of training? Is it decided by researchers or participants? As it’s not a routine scenario, to make a fair comparison with machine vision, it should be decided by participants and an additional test should be added to validate that participants have been well-trained.

The procedure was as follows: We first verbally explained the type of actions to the participant with the help of wooden cubes, and then we asked the participant to put the VR headset on his/her eyes and watch these explained actions in the world of virtual reality. For each action, we showed a sample and the name of the action appeared in a green box in front of the participant throughout the action. After showing one experimental sample for each of the ten actions, we asked the participant if everything was clear, and if he/she confirmed, we would start the experiment. Now this is written in text too (see lines 167-169)

Regarding the training level, is it important to note that the ten actions we used were very simple and are integral part of everyday life object manipulation. Prediction accuracy during the experiment was very high with a mean of 97.6%, which underpins that the task was well understood. 

6) Are participants informed that their response time will be recorded as an evaluation criterion, which may affect their prediction timing?

Yes, all the details including the importance of response time were explained to each participant before the experiment. The instruction was “indicate as quickly as possible, which action was currently presented”, but we did not ask the participant to do this in any competitive way (e.g. competing against other participants or the machine), avoiding this way time pressure. Thus, we used an absolutely conventional approach to instructing participants in a behavioral experiment where error rates and reaction times are recorded. 

7) What about prediction accuracy? Are all prediction results correct? How to analyze wrong predictions?

We thank the reviewer for this important inquiry. We added the prediction accuracy to the results section. Participants' mean prediction accuracy was very high (M = 97.6%, SD = 1.8, n = 49). Therefore, we did not further analyze wrong predictions (apart from the correlational analysis of the error rate to identify learning effects) (see lines 318-319 and 322-325).

Please, note that we detected a coding mistake in our trial variable, which we now corrected. This led to changes in the results section concerning the learning effects - now showing a significant reduction in error rate and enhancement in predictive power in the course of the experiment, which is fully in line with what one would expect for human behavior (see lines 321-324).

8) A typo mistake in line 140: two “for example”.

Thank you - now corrected

*****Reviewer 2*****

This paper proposes a system about machine based action recognition system eSEC learning and designed a virtual reality setup and tested recognition speed for different manipulation actions. The authors introduce in details how the theoretical analysis is done and recognition speed is performed.

Paper is not well organized and has limited potential for acceptance in “PLOS ONE”, in current format though there are some observations, corrections and suggestions regarding this paper.

1) Author MUST clearly describe their contribution. Put another section what is author contribution?

Now we bullet list our contributions in the introduction (see lines 107-122): 

“The current study consisted of the following steps:

• Creating a virtual reality database containing ten different manipulation actions with multiple scenarios each.

• Conducting a behavioural experiment in which human participants engaged in action prediction in virtual reality for all scenarios, where prediction time and prediction accuracy were measured.

• Calculating three types of spatial relations using the eSEC model: (1) touching vs. non-touching relations, (2) static spatial relations and (3) dynamic spatial relations.

• Performing an information theoretical analysis to determine how participants used these three types of spatial relations for action prediction.

• Training an optimal (up to the learning accuracy) machine algorithm to predict an action using the relational information provided by the eSEC model.

• Comparing human to the optimal machine action prediction strategies based on spatial relations.”

2) Separate introduction and literature review.

To merge the introduction with the literature review is standard in psychology papers. Thus, in this multidisciplinary paper we leave it this way, especially in view that this paper focuses onto the investigation of human action prediction with machine prediction providing the point of comparison for the analysis of human performance. 

3) Proposed work section is quite weak and needs major improvement. It lacks any flow diagram, algorithm, pseudo code etc. Each step of proposed algorithm/work should be clearly depicted how your work is different from existing work.

We have updated the flow diagram presented in Figure 1, which now better explains of which parts our study is composed. We are providing descriptions for each block of that flow diagram in the Method section. However, in the main text we are providing only the essential (intuitive) description of the main algorithmic steps of the eSEC model and we are providing all finer computational details in the Appendix. This is done with the aim not to over-burden the main text with computational details and make it accessible to readers interested in the psychological aspects of the study, which are our core contribution.

The differences from existing work are described in the introduction and discussion sections. Essentially, we investigate for the first time how humans recognize actions based on relational spatial information between manipulated objects, when context is not available. 

Here we summarize what is old and what is new with respect to the methods in our study. Old methods: we use the eSEC model developed in our previous work and we use standard statistical methods to investigate the obtained human data. New is the virtual reality setup in which we perform our experiments and the virtual reality database with 300 manipulation action scenarios (10 actions, 30 scenarios each) which we demonstrate to the participants of our study. 

4) Diagrams and flow charts are not good need to redraw.

Thank you for drawing our attention to this shortcoming. All figures were reproduced using higher resolution (600 dpi). 

Please, note, that the figures in the merged submission PDF are not reproduced with good quality, but the PLOS guarantees the download of the high-resolution figures, see https://everyone.plos.org/2011/02/11/ask-everyone-figure-files-in-your-merged-pdf/

5) Performance measures should be more. The proposed work should be evaluated with a number of performance measures to prove its validity.

Thank you for this remark. We added the human prediction accuracy as performance measure. 

All in all, in respect to methods, we now did the following:

• We performed a repeated measures ANOVA to analyze human predictive power and compared human and machine predictive power for different actions using t-tests.

• We calculated the information gain based on each eSEC column entry, and fitted logistic regression to the obtained series for eight different sets of spatial relations (Touching, Static and Dynamic alone, as well as all possible combinations of those types of relations plus one model which does not divide the relations into separate components). 

• We analyzed learning effects in human error rates and predictive power.

We would argue that this is a set of methods, both representative and allowing us achieving trustworthy results, showing that humans successfully exploit relational spatial information for action prediction. Moreover, the set of methods allows us to determine what strategies humans deploy when provided only with dynamic and static spatial information for action prediction.

6) Abstract and Conclusion are poorly written need much revision.

Abstract and discussion sections have been rewritten. Conclusion section was incorporated into the discussion.

7) Add references that are more recent.

We added new references in the introduction: for definition of more recent psychological findings and for recent machine learning aspects (Stadler et al, 2012, Ref. No. [12], Wurm et al, 2014, Ref. No. [14], Cheng et al. 2020, Ref. No. [35], Ejdeholm et al. 2020, Ref. No. [40]), as well as Barros et al, 2017, Ref. No. [48] and Sun et al, 2019, Ref. No. [49] in the discussion. 

8) Overall, the paper lies in the category of revision.

Thanks for that opportunity! We hope that the revised manuscript is much clearer now.

9) Overall the language is not very good; however, it MUST be proofread if again before submission again.

We have proof-read the paper.

*****Reviewer 3*****

Action prediction independent of object information as an observation and hypothesis is validated through a psycho-physical experiments rigorously conducted by the authors. A set of 10 actions are considered over a VR based experimental system. The authors further validated an eSEC computational framework to show that with eSEC, machine could achieve action prediction capability. The machine prediction power vs human prediction performance as a comparison is provided by the authors and some speculative explanations are given and discussed.

1) Some parts of the draft could be improved by making the description more clear to the readers. For example, line 396 "when all three types of information", It is unclear to me what is the third type of information other than dynamic and static ones? Please clarify. 

Now we write (see lines 376-377): “Notably, when all three types of information (i.e., touching, static or dynamic information) were equally beneficial (this was the case for take, uncover, shake, and put)…”

We also name the three types of information explicitly in the abstract, introduction, method and result sections to make the writing clearer.

2) Also, an interesting future question and direction could be, as most of the action recognition dataset and benchmarks in computer vision research area come up with the set of actions in a kind of ad-hoc manner (especially for manipulation action dataset). I would be keen to see the authors based on their discoveries from this draft to provide some designing principles for future action recognition dataset and challenges that could fully consider the types of information discussed here.

Possibly the most interesting aspect for any new data set relates to the finding that we can predict actions often very efficiently in situations, where context is not interpretable (such as those introduced by our VR cube-world here). This suggests creating such a data set which should also include actions, where objects are used in unconventional ways (like cutting dough with a spoon, etc.), to avoid fully object-based action recognition strategies. Comparing results from such data with results from more conventional (object- or context oriented) data should allow us to better understand the role and affordances of objects. This, however, can only be done in future work.

*****Reviewer 4*****

1) Abstract is too much lengthy.

The Abstract has been sharpened, but – to address the next comment of the reviewer – methods, etc. had been clearly pointed out in the Abstract, too, leading to the fact that not much of its length could be reduced.

2) Actual methodology/algorithms are not mentioned in the abstract.

The eSEC model with some details and information theoretical analysis, the two stepping-stones of our methodological approach, are mentioned in the abstract.

3) In introduction section, contributions should be mentioned in bullets for better understanding of the readers.

In the revised manuscript, we added a bullet list of contributions. Now we write (see lines 107-122):

 “The current study consisted of the following steps:

• Creating a virtual reality database containing ten different manipulation actions with multiple scenarios each.

• Conducting a behavioural experiment in which human participants engaged in action prediction in virtual reality for all scenarios, where prediction time and prediction accuracy were measured.

• Calculating three types of spatial relations using the eSEC model: (1) touching vs. non-touching relations, (2) static spatial relations and (3) dynamic spatial relations.

• Performing an information theoretical analysis to determine how participants used these three types of spatial relations for action prediction.

• Training an optimal (up to the learning accuracy) machine algorithm to predict an action using the relational information provided by the eSEC model.

• Comparing human to the optimal machine action prediction strategies based on spatial relations.”

4) The manuscript should be checked for typos. In some places the word Figure is written while in other places Fig is written, must be uniform throughout.

We use only “Figure” now. We also have proof-read the entire manuscript. 

5) Figures quality is not good, must be 300dpi.

We reproduced all figures at 600 dpi.

Please, note, that the figures in the merged submission PDF are not reproduced with good quality, but the PLOS guarantees the download of the high-resolution figures, see https://everyone.plos.org/2011/02/11/ask-everyone-figure-files-in-your-merged-pdf/

6) Authors employed so-called extended semantic event chains (eSEC) which is an existing work, what is their real contribution?

The motivation of the study was to find out how humans use spatial relations between objects for prediction of manipulation actions using the eSEC predictions as a point of comparison. Our study is the first to investigate how humans recognize actions based on relational spatial information between manipulated objects, when contextual information is not available.

For a full list of contributions see our bullet list in lines 108-122.

---

## [Decision Letter · Decision Letter 1]

27 Nov 2020

Using enriched Semantic Event Chains to model human action prediction based on (minimal) spatial information

PONE-D-20-16126R1

Dear Dr. Fatemeh Ziaeetabar,

We’re pleased to inform you that your manuscript has been judged scientifically suitable for publication and will be formally accepted for publication once it meets all outstanding technical requirements.

Kind regards,

Chen Zonghai

Academic Editor

PLOS ONE

Additional Editor Comments (optional):

Based on the opinions of the reviewers, it is suggested that the manuscript be accepted.

Reviewers' comments:

Reviewer's Responses to Questions

**Comments to the Author**

1. If the authors have adequately addressed your comments raised in a previous round of review and you feel that this manuscript is now acceptable for publication, you may indicate that here to bypass the “Comments to the Author” section, enter your conflict of interest statement in the “Confidential to Editor” section, and submit your "Accept" recommendation.

Reviewer #1: All comments have been addressed

Reviewer #4: All comments have been addressed

2. Is the manuscript technically sound, and do the data support the conclusions?

Reviewer #1: Yes

Reviewer #4: Yes

3. Has the statistical analysis been performed appropriately and rigorously? 

Reviewer #1: Yes

Reviewer #4: Yes

4. Have the authors made all data underlying the findings in their manuscript fully available?

Reviewer #1: Yes

Reviewer #4: Yes

5. Is the manuscript presented in an intelligible fashion and written in standard English?

Reviewer #1: Yes

Reviewer #4: Yes

6. Review Comments to the Author

Reviewer #1: The author made necessary changes to the manuscript（ PONE-D-20-16126） and answered the concerns of the reviewer.

Reviewer #4: Authors addressed all comments, it is accepted in its current form. It is also recommended for publication

7. PLOS authors have the option to publish the peer review history of their article (what does this mean?). If published, this will include your full peer review and any attached files.

Reviewer #1: No

Reviewer #4: No

---

## [Editor Report · Acceptance letter]

14 Dec 2020

PONE-D-20-16126R1 

Using enriched Semantic Event Chains to model human action prediction based on (minimal) spatial information 

Dear Dr. Ziaeetabar:

I'm pleased to inform you that your manuscript has been deemed suitable for publication in PLOS ONE. Congratulations! Your manuscript is now with our production department. 

Kind regards, 

on behalf of

Prof. Chen Zonghai 

Academic Editor

PLOS ONE